# Elevated Plasma Levels of C1qTNF1 Protein in Patients with Age-Related Macular Degeneration and Glucose Disturbances

**DOI:** 10.3390/jcm11154391

**Published:** 2022-07-28

**Authors:** Agnieszka Budnik, Marta Sabasińska-Grześ, Magdalena Michnowska-Kobylińska, Łukasz Lisowski, Małgorzata Szpakowicz, Magdalena Łapińska, Anna Szpakowicz, Marcin Kondraciuk, Karol Adam Kamiński, Joanna Konopińska

**Affiliations:** 1Department of Ophthalmology, Medical University of Białystok, 15-089 Bialystok, Poland; agnieszka-bartczak@o2.pl (A.B.); m.sabasinska@vp.pl (M.S.-G.); magdalena.michnowska.evita@gmail.com (M.M.-K.); lisowski@vp.pl (Ł.L.); 2Department of Population Medicine and Lifestyle Diseases Prevention, Medical University of Białystok, 15-259 Bialystok, Poland; malgorzata.szpakowicz@umb.edu.pl (M.S.); magdalena.lapinska@umb.edu.pl (M.Ł.); marcin.kondraciuk@umb.edu.pl (M.K.); karol.kaminski@umb.edu.pl (K.A.K.); 3Department of Cardiology, Medical University of Bialystok, 15-089 Bialystok, Poland; anna.szpakowicz@umb.edu.pl

**Keywords:** AMD, age-related macular degeneration, C1qTNF, glucose disturbances

## Abstract

In recent years, research has provided increasing evidence for the importance of inflammatory etiology in age-related macular degeneration (AMD) pathogenesis. This study assessed the profile of inflammatory cytokines in the serum of patients with AMD and coexisting glucose disturbances (GD). This prospective population-based cohort study addressed the determinants and occurrence of cardiovascular, neurological, ophthalmic, psychiatric, and endocrine diseases in residents of Bialystok, Poland. To make the group homogenous in terms of inflammatory markers, we analyzed only subjects with glucose disturbances (GD: diabetes or prediabetes). Four hundred fifty-six patients aged 50–80 were included. In the group of patients without macular degenerative changes, those with GD accounted for 71.7%, while among those with AMD, GD accounted for 89.45%. Increased serum levels of proinflammatory cytokines were observed in both AMD and GD groups. C1qTNF1 concentration was statistically significantly higher in the group of patients with AMD, with comparable levels of concentrations of other proinflammatory cytokines. C1qTNF1 may act as a key mediator in the integration of lipid metabolism and inflammatory responses in macrophages. Moreover, C1qTNF1 levels are increased after exposure to oxidized low-density lipoprotein (oxLDL), which plays a key role in atherosclerotic plaque formation and is also a major component of the drusen observed in AMD. C1qTNF1 may, therefore, prove to be a link between the accumulation of oxLDL and the induction of local inflammation in the development of AMD with concomitant GD.

## 1. Introduction

Age-related macular degeneration (AMD) is a leading cause of irreversible blindness in developed countries. It is estimated that its prevalence will rise to affect approximately 288 million people worldwide by 2040 [1]. The etiology of AMD is complex, and its risk factors for onset and progression include age (≥50 years old), ocular dysfunction, systemic diseases, diet, smoking, environmental, behavioral, and genetic factors [2,3,4]. The exact pathophysiology of AMD is not fully understood. The changes that occur during its course lead to irreversible vision loss, often preventing patients from engaging in professional activities and significantly altering everyday functioning. This, in itself, is the reason it is of paramount importance to investigate additional therapeutic methods of preventing and limiting its course.

The theories of dysregulation in the complement, lipid, angiogenic, inflammatory, and extracellular matrix pathways are all presently being considered in its pathogenesis [2]. In recent years, research has provided increased evidence for the importance of inflammatory etiology in AMD pathogenesis [5].

The crucial role of maintaining homeostasis in photoreceptor cells is played by the retinal pigment epithelium (RPE), a monolayer of hexanocuboidal pigmented epithelial cells located between the neurosensory retina and the choroid [6]. With age, the metabolic and phagocytic function of RPE cells decreases, leading to the focal accumulation of lipids, proteins, and minerals [7]. In combination with changes in the permeability of Bruch’s membrane, this causes the deposition of drusen between the RPE and Bruch’s membrane. Drusen—apart from lipoproteins, polysaccharides, and glycosaminoglycans—contains inflammatory factors, and complements system proteins and components of dendritic cell processes, which supports immunity in their genesis [8,9,10]. Drusen most likely promote immune system activation by initiating inflammation and oxidative stress [11].

There is more evidence for the involvement of immune cells in neovascular AMD (nAMD) as well. Increased levels of proinflammatory cytokines in the aqueous humor [12] and the activation of the complement system in serum have been demonstrated in patients with nAMD [13,14]. The damaged RPE produces VEGF, interleukin-8 (IL-8), and monocyte chemoattractant protein-1 (MCP), which attract monocytes from the choriocapillaris along the outer surface of Bruch’s membrane. The released inflammatory cytokines and VEGF result in a breakdown of Bruch’s membrane, leading to the growth of abnormal choroidal vessels toward the retina. Histological research has shown the presence of macrophages within the choroidal neovascularization (CNV), and that the amount present is correlated with the thickness of the lesions and the degree of fibrosis. The macrophages express TNF-α and interleukin-1(IL-1), which up-regulate complement factor-B, activate the complementary alternative pathway in the subretinal space, and stimulate RPE cells to produce even more VEGF. IL-6, the pro-inflammatory cytokine that plays an important role in the development of CNV, was found to be increased in the serum and eye aqueous humor of patients with neovascular AMD.

In addition, several studies have shown that diabetes is a risk factor for AMD [15,16,17]. The exact mechanisms have yet to be established; however, it is thought that oxidative stress and inflammation may play a role [18,19,20]. Hyperglycemia and dyslipidemia in diabetic patients interrupt homeostasis of the retina by provoking inflammatory responses in RPE, including oxidative stress [20]. Substantially elevated oxidative stress markers were found in the serum of AMD patients compared to healthy controls [21]. Inflammation disrupts the signaling pathways of many inflammatory cytokines and chemokines, including those involving the tumor necrosis factor (TNF), interleukin-6 (IL-6), IL-1b, C-reactive protein (CRP), CC-chemokine ligand 2 (CCL2), and adipokines. These inflammatory activators promote the dysfunction and apoptosis of the RPE and photoreceptor cells [22]. Diabetes may lead to an accumulation of the highly stable advanced glycation end products (AGEs) in RPE cell layers and photoreceptors. AGEs have been reported to accumulate in AMD patients’ eyes in Bruch’s membrane, drusen, subfoveal neovascular membranes, and RPE cells [23].

The objective of this study was to assess the profile of inflammatory cytokines in the serum of patients with AMD and coexisting glucose disturbances in a population study and to establish methods for differentiating between its profile within AMD and non-AMD patients.

## 2. Methods

### 2.1. Bialystok PLUS Study

The participants were aged 50–80 years and were derived from the Bialystok PLUS study. This is a prospective population-based cohort study that addresses determinants and occurrence of cardiovascular, neurological, ophthalmic, psychiatric, and endocrine diseases in residents in Bialystok, Poland. The aims and design of the Bialystok PLUS study have been described in detail elsewhere [24]. Data were collected from 11/2018 to 06/2021. The lower age limit of 50 years was applied because this is one of the criteria used to diagnose AMD.

To make the group homogenous in terms of inflammatory markers, we analyzed only subjects with diabetes or prediabetes (glucose disturbances). Normal glucose metabolism was defined as: fasting glucose < 100 mg/dL and 2 h 75 g oral glucose tolerance test (OGTT) < 140 mg/dL. Glucose disturbance was defined as: (1) history of diabetes, (2) patients newly diagnosed with diabetes based on OGTT, (3) impaired fasting glucose (≥100 mg/dL), (4) impaired glucose tolerance (glucose after 2 h in OGTT ≥ 140 mg/dL). History of diabetes was derived from computerized baseline questionnaires. Routine biochemical tests were assessed using a Cobas c111 machine (Roche, Basel, Switzerland). Blood morphology was studied using a Mythic 18 machine (Cormay, Warsaw, Poland). Proteomic analysis was performed with Olink inflammation panel (Olink Proteomics AB, Upssala, Sweden). Body mass index (BMI) was calculated as weight in kilograms divided by height in meters squared.

### 2.2. Diagnosis of AMD

All participants underwent fundus photography without pharmacologic mydriasis with a 35° digital color fundus camera (Canon CR-2 PLUS AF, New York, NY, USA). Fundus pictures were graded according to the Wisconsin age-related maculopathy grading system [25] and the modified International Classification System [26] by trained retinal specialists (ŁL, MMK). The eyes of each participant were graded and classified separately, and the eye with the more severe grade was used to classify the person. Following a previous population study [27], we included early and late AMD in analysis. These lesions included retinal pigmentary alterations, large drusen (≥125 μm), and large drusen area (≥5,331,820 μm^2^) [19]. In the manuscript, this is all referred to as AMD.

### 2.3. Statistical Analysis

The categorical variables were compared with chi-square tests. Continuous variables with normal distributions were compared using Student’s t-test (with or without Welch’s correction). Continuous variables with non-normal distributions were compared using the Wilcoxon signed-rank test for independent groups.

The study was conducted according to the Declaration of Helsinki and all participants gave written informed consent. The ethical approval for this study was provided by the Ethics Committee of the Medical University of Bialystok (Poland) on 31 March 2016 (approval number: R-I-002/108/2016).

## 3. Results

The sample was comprised of 456 patients aged 50–80 years old. A total of 20 patients were excluded from the study due to incomplete examination. AMD was diagnosed in 47 subjects (10.8% of the total sample). In the group of patients without macular degenerative changes, those with glucose disturbances accounted for 71.7%, while among those with AMD, diabetics accounted for 89.4%. The difference was statistically significant (*p* = 0.016).

Demographic characteristics of the sample are shown in Table 1.

Further analyses were performed among 321 patients with glucose disturbances. Our study groups were homogeneous with respect to the distribution of sex and age. Subjects from the AMD 0 and AMD 1 groups did not exhibit any statistically significant differences in the mean value of HbA1C. BMI value was statistically significantly higher in the AMD 0 group than in the AMD 1 group (*p* = 0.035; see Table 2).

Table 3 and Appendix A (proteomic analysis) present levels of parameters associated with inflammatory action based on AMD status. A statistically significant difference between the groups was observed in the case of the C1qTNF1 protein; patients with AMD had a significantly higher expression than the healthy control group (5.14 ± 0.34 NPX [normalized protein expression] vs. 4.91 ± 0.38 NPX, *p* = 0.02.

## 4. Discussion

Increased serum levels of proinflammatory cytokines were observed in both AMD and diabetes patients [28,29,30,31]. We found that the level of C1qTNF1 concentration is different in patients with glucose disturbances and AMD versus patients with glucose disturbances without macular degenerative changes. The concentration was statistically significantly higher in the group of patients with AMD, with comparable concentration levels of other proinflammatory cytokines and VEGFA. C1qTNF1 belongs to a family of secreted proteins that are homologous to adiponectin, called C1q/TNF-related proteins (CTRPs).

CTRPs are composed of four different domains: an N-terminus signal peptide, a short variable region, a collagenous domain, and a C-terminal globular domain. The globular domain shows structural similarity with the C1 q complement component and TNF ligand family, from which the CTRP family takes its name. CTRP group proteins are found in many tissues of the body. CTRP1 is expressed in the heart, placenta, liver, kidney, muscle, prostate, ovary, adipose tissue, and stromal vascular cells [32]. CTRP1 has many important functions in the body. It affects glucose metabolism by lowering blood glucose concentrations and increasing tissue insulin sensitivity. It also plays a role in reducing ischemia-reperfusion damage in myocardial infarction. It has antithrombotic effects through the inhibition of von Willebrand factor34. CTRP1 affects chondrocyte proliferation and maturation and aldosterone synthesis. It plays a role in promoting the oncogenic process. Disturbed expression of CTRP1 is observed in hepatocellular carcinoma [33], glioblastoma [34], and osteosarcoma [35].

The serum levels of CTRP proteins are affected by age, gender, and obesity [36]. The level of CTRP1 is higher in subjects with type 2 diabetes mellitus. Our study groups were homogeneous with respect to the distribution of sex, age, and HbA1C levels, which excludes the influence of the above factors on serum CTRP1 levels.

In a study by Zhu et al., CTRP1 was shown to increase vascular endothelial cell permeability via VEGFR receptor 2 activation [37]. Increased endothelial permeability has been found in mice with genetically engineered CTRP1 overexpression, and reduced permeability has been found in animals with CTRP1 deletion. In addition, increased phosphorylation of VEGFR2 has been demonstrated under treatment with recombinant CTRP1, which was blocked after the administration of specific CTRP1 neutralizing antibodies. This study demonstrated increased permeability of Evans Blue Dye under CTRP1 in various body organs, indicating that CTRP1 controls endothelial cell permeability in both large and small vessels.

The effect of CTRP1 on vascular endothelial cells via VEGFR2 signaling is of particular interest with regard to the retina. VEGFR2 belongs to a group of VEGF-related receptors that includes two other receptors, VEGFR1 and VEGFR3, all of which are composed of an extracellular ligand-binding domain, transmembrane domain, juxtamembrane domain, and intracellular domain. VEGF receptors differ in their ligand-binding properties and their function in the body. VEGFR2 is found in vascular endothelial cells but is also expressed in some neuronal cells in the retina [38]. It is mainly through VEGFR2 that VEGF exerts its mitogenic, angiogenic, and permeability-enhancing effects [39]. VEGFR2 plays a key role in the pathogenesis of the exudative form of AMD. In recent years, however, evidence is emerging supporting the importance of VEGFR2 in the development of the dry form of AMD as well. In a study by Bohan Xu et al., it was found that VEGFR2 levels are a determinant of RPE cell death under glucose deprivation [40].

Currently used drugs block the action of VEGFA. This achieves the therapeutic goal of reducing retinal edema, but it has its side effects. It has been observed that long-term use of intravitreal anti-VEGF injections increases the risk of geographic atrophy. Current research is focused on finding methods to selectively block the VEGFR2 receptor to attenuate the retinal edema caused by its activation while avoiding the adverse effects of its blockade in the form of vascular rarefaction and geographic atrophy [41].

In our study, patients with AMD did not differ in serum VEGFA levels from patients without macular degenerative changes. Thus, it is important to consider that it may be that elevated CTRP1 levels, through VEGFR2 receptor activation, contributed to the development of AMD-specific macular changes.

Many researchers emphasize that the drusen in AMD resemble atherosclerotic plaques, extracellular deposits seen in elastosis and amyloidosis, or plaques in Alzheimer’s disease [42]. The local inflammation that stimulates the formation of drusen resembles the processes that occur in the diseases mentioned above [42]. There is a growing hypothesis that AMD can be a local manifestation of systemic disease [13]. The link between atherosclerosis and AMD is evidenced by epidemiological data, the presence of the same proteins in the drusen and plaque, and the structural commonalities between the vessel wall and Bruch’s membrane [43]. CTRP1 promotes atherosclerotic plaque development by promoting the production of proinflammatory cytokines and leukocyte-endothelial interactions. Results suggest that CTRP1 may act as a key mediator in the integration of lipid metabolism and inflammatory responses in macrophages. It has been observed that CTRP 1 levels are increased after exposure to oxLDL, which plays a key role in atherosclerotic plaque formation. In the development of atherosclerosis, monocytes are transformed into macrophages, which turn into foam cells after the internalization of oxLDL. CTRP1 production has been shown to significantly increase in lipid-laden foam cells. The internalization of oxLDL not only leads to lipid accumulation in the arterial wall but also enhances local inflammation. This probably occurs via CTRP1, which exerts its pro-inflammatory effects via autocrine/paracrine mechanisms. OxLDL is also a major component of the drusen observed in AMD [44]. Accumulation of OxLDL is believed to contribute to the pathogenesis of AMD by establishing a low-grade, chronic inflammatory state in the macula. The exact mechanism linking OxLDL accumulation to the development of AMD is poorly understood. Given that CTRP1 levels are elevated in patients with AMD, it is worth considering whether it also acts as a proinflammatory in the drusen under the influence of OxLDL as it does in the atherosclerotic plaque. CTRP1 may prove to be a link between the accumulation of OxLDL and the induction of local inflammation and the development of AMD and concomitant GD.

The usefulness of our study is limited by its observational character, the relatively small number of patients, and the lack of detailed data on AMD status. To the best of our knowledge, this is the first publication showing elevated CTRP1 levels in AMD patients with concomitant glucose disturbances. CTRP1 may be involved in the development of degenerative changes in the macula. Future studies are needed to explore the precise role of CTRP1 in the pathogenesis of AMD, which may result in the discovery of novel treatment targets. It may also play a role in the early diagnosis of AMD in patients with concomitant GD.

## Figures and Tables

**Table 1 jcm-11-04391-t001:** Demographic characteristics of the sample by AMD diagnosis.

	AMD 0 *n* = 389	AMD 1 *n* = 47	MD/OR 95% CI	*p*
Sex (W), *n* (%)	162 (41.6)	22 (46.8)	0.81 (0.44; 1.49)	0.498 ^1^
Age, Me (Q1; Q3)	63.00 (58.00; 68.00)	66.00 (59.00; 69.00)	−3.00 (−4.00; 0.00)	0.100
DM or prediabetes *n* (%)	279 (71.7)	42 (89.4)	3.31 (1.28; 8.59)	0.016

Q1/Q3—Quartile 1 and 3. The relationships between sex, diabetes, and the group were analyzed with chi-square tests ^1^. The Wilcoxon test for independent groups was used to compare ages between the groups. MD/OR 95% CI—median difference (difference between medians (AMD 0 − AMD 1))/odds ratio with 95% confidence intervals.

**Table 2 jcm-11-04391-t002:** Characteristics of patients with glucose disturbances by AMD diagnosis.

Variables	AMD 0 (*n* = 279)	AMD 1 (*n* = 42)	MD/OR 95% CI	*p*
Sex (W), *n* (%)	130 (46.6)	20 (47.6)	0.96 (0.50; 1.84)	>0.999
Age, Me (Q1; Q3)	63.00 (59.00; 69.00)	65.50 (57.00; 69.00)	−2.50 (−3.00; 2.00)	0.455 ^1^
HBA1C, Me (Q1; Q3)	5.70 (5.40; 6.00)	5.70 (5.40; 6.08)	0.00 (−0.20; 0.10)	0.860 ^1^
BMI, Me (Q1; Q3)	29.03 (25.81; 32.56)	28.21 (23.92; 31.47)	0.82 (0.11; 3.48)	0.035 ^1^

Q1/Q3—Quartile 1 and 3. The relationship between sex and group designation was analyzed using the chi-square test. The Wilcoxon ^1^ test for independent groups was used to compare age, BMI, and HBA1C. MD/OR 95% CI—median difference (difference between medians (AMD 0 − AMD 1))/odds ratio with 95% confidence intervals.

**Table 3 jcm-11-04391-t003:** Characteristics and comparisons of AMD 0 and AMD 1 groups.

Variables	AMD 0	AMD 1	MD/OR 95% CI	*p*
	M ± SD (Q1; Q3)			
	*n* = 279	*n* = 42		
WBC (10^3^/µL)	6.15 (5.30; 7.20)	6.60 (5.45; 7.70)	−0.45 (−0.90; 0.10)	0.131
LYM (10^3^/µL)	1.85 (1.53; 2.28)	2.00 (1.70; 2.20)	0.15 (−0.30; 0.10)	0.312
MONO (10^3^/µL)	0.30 (0.30; 0.40)	0.35 (0.30; 0.40)	0.05 (0.00; 0.00)	0.858
NEUT (10^3^/µL)	3.80 (3.20; 4.60)	4.30 (3.00; 5.10)	−0.50 (−0.70; 0.20)	0.308
LYM (%)	31.05 ± 7.02	30.99 ± 8.86	0.06 (−2.82; 2.93)	0.967 ^1^
MONO (%)	5.40 (4.60; 6.50)	5.25 (4.30; 6.05)	0.15 (−0.10; 0.70)	0.199
NEUT (%)	63.50 (58.55; 67.90)	63.95 (56.25; 70.95)	−0.45 (−3.90; 1.90)	0.513
HsCRP (mg/L)	1.03 (0.53; 2.18)	0.95 (0.31; 1.44)	0.08 (−0.01; 0.54)	0.058
Ferritin (ng/mL)	168.40 (108.60; 278.38)	128.15 (85.59; 204.55)	40.25 (−2.80; 69.90)	0.070
IL-6 (pg/mL)	3.16 (2.17; 4.33)	2.90 (2.07; 4.04)	0.26 (−0.25; 0.98)	0.323
C1qTNF1 (NPX)	4.91 ± 0.38	5.14 ± 0.34	−0.23 (−0.43; −0.03)	0.022 ^1^
VEGF-A (NPX)	12.05 ± 0.54	12.05 ± 0.59	0.00 (−0.29; 0.30)	0.983 ^1^

Q1/Q3—Quartile 1 and 3. Student’s t-test was used for between-groups comparisons of variables with normal distributions ^1^ (with or without Welch’s correction). Variables without a normal distribution were compared using Wilcoxon’s test for independent samples. MD 95% CI—mean/median difference (AMD 0–AMD 1) with 95% confidence intervals.

## Data Availability

All materials and information will be available upon an e-mail request to the corresponding author.

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
