# Peer review of "Elevated Plasma Levels of C1qTNF1 Protein in Patients with Age-Related Macular Degeneration and Glucose Disturbances"

_jcm, 2022, doi:10.3390/jcm11154391_

Round 1

Reviewer 1 Report

The authors present interesting and novel finding regarding possible AMD mechanism, however only in patients with GD, which may not apply for AMD patients in general. I would suggest, that authors must emphasize this more in the paper in general (e.g. add ''and comcomitant GD'' to line 27-28)  

In methods section authors say'Besides AMD 117 we also investigated AMD-specific lesions as a separate outcome variable. These lesions 118 included retinal pigmentary alterations, large drusen (≥125 μm), and large drusen area 119 (≥5,331,820 μm2) '

What separate outcomes do you mean? This was not described in the result section.

Table 1 include 436 patients in total, while the authors state, that 456 patients were included. What are the reasons not to include these 20 patients ?

Author Response

Reviewer 1

Comment 1

The authors present interesting and novel finding regarding possible AMD mechanism, however only in patients with GD, which may not apply for AMD patients in general. I would suggest, that authors must emphasize this more in the paper in general (e.g. add ''and concomitant GD'' to line 27-28)  

Response

We are grateful to the Reviewer for his insightful comments on our
manuscript. We have incorporated changes to reflect all of the suggestions provided by the Reviewer. We have added the phrase ''with concomitant GD'' inline 27-28.

Comment 2

In methods section authors say'Besides AMD 117 we also investigated AMD-specific lesions as a separate outcome variable. These lesions 118 included retinal pigmentary alterations, large drusen (≥125 μm), and large drusen area 119 (≥5,331,820 μm2) '

What separate outcomes do you mean? This was not described in the result section.

Response

We are sorry for translating error and misleading .We meant that: “Following previous population study[i] we included in the analysis’s incident early and late AMD. These lesions included retinal pigmentary alterations, large drusen (≥125 μm), and large drusen area (≥5,331,820 μm2) (20). In the manuscript this is all referred to as AMD.” The change was made accordingly.

Comment 3

Table 1 include 436 patients in total, while the authors state, that 456 patients were included. What are the reasons not to include these 20 patients ?

Response

20 patients were excluded from the study due to incomplete examination. The clarification has been added. (line 132)

Reviewer 2 Report

The research is well conducted and the article is well written, I would suggest to the Authors to improve the discussion about the practical effect that their research has on ophthalmological practice

I would suggest the following minor revision: - References are written in different format : e.g line 34 is [1] while line 35 is 1. Please fix it in the whole text

- Moreover references should be cited before the final point of a sentence

- Line 44 please provide at least one reference for the following statement “In recent 44 years research has provided increasing evidence for the importance of inflammatory etiology in AMD pathogenesis”

- The article is well written, English does not need editing - Statistics is well done

- The only point I would address is : how does this research translate into clinical practice? Is there any practical application?

Author Response

Reviewer 2

Comment 1

The research is well conducted and the article is well written, I would suggest to the Authors to improve the discussion about the practical effect that their research has on ophthalmological practice

Response

We are grateful to the Reviewer for his insightful comments on our
manuscript. We have added information about potential use of the study in ophthalmological practice. (lines 245-254)

Comment 2

I would suggest the following minor revision: - References are written in different format : e.g line 34 is [1] while line 35 is 1. Please fix it in the whole text

Response

Thank you for that comment. We have fixed it in whole text.

Comment 3

- Moreover references should be cited before the final point of a sentence

Response

Thank you for that comment. We have made the change accordingly.

Comment 4

- Line 44 please provide at least one reference for the following statement “In recent 44 years research has provided increasing evidence for the importance of inflammatory etiology in AMD pathogenesis”

Response

Thank you for that comment. We have added missing reference: [1] Datta S, Cano M, Ebrahimi K. et al. The impact of oxidative stress and inflammation on RPE degeneration in non-neovascular AMD. Prog Retin Eye Res. 2017;60:201–18.

Comment 5

- The article is well written, English does not need editing - Statistics is well done

Response

Thank you for that comment. We appreciate it!

Comment 6

- The only point I would address is : how does this research translate into clinical practice? Is there any practical application?

Response

We have added information about potential use of the study in ophthalmological practice (lines 249-252): The future studies are needed to explore the precise role of CTRP1 in pathogenesis of AMD, that can result in discover novel treatment target. It may also play a role in early diagnosis of AMD in patients with concomitant GD.

Reviewer 3 Report

Overall a decent job. I had a few minor quibbles with the English which were marked but nothing of significance to understanding the paper. I thought the discussion was in depth and excellent in quality. The discussion needs to address a couple of points which I made notes on. I think a few standard comments on weaknesses need to be added to the discussion as basically all papers have flawes and weaknesses inherent to their design.  Awareness of them is important to both readers and authors.

Author Response

Reviewer 3

 Overall a decent job. I had a few minor quibbles with the English which were marked but nothing of significance to understanding the paper. I thought the discussion was in depth and excellent in quality. The discussion needs to address a couple of points which I made notes on. I think a few standard comments on weaknesses need to be added to the discussion as basically all papers have flawes and weaknesses inherent to their design.  Awareness of them is important to both readers and authors.

Response

We are grateful to the Reviewer for his insightful comments on our
manuscript. We have incorporated changes to reflect all of the suggestions provided by the Reviewer. We have added information and improved discussion as well as removed all spelling errors for which we apologies. Moreover we have added the sentence about limitations of the study.